# Differential Accumulation of Metabolites and Transcripts Related to Flavonoid, Styrylpyrone, and Galactolipid Biosynthesis in *Equisetum* Species and Tissue Types

**DOI:** 10.3390/metabo12050403

**Published:** 2022-04-29

**Authors:** Amber N. Parrish, Iris Lange, Dunja Šamec, Bernd Markus Lange

**Affiliations:** Institute of Biological Chemistry and M.J. Murdock Metabolomics Laboratory, Washington State University, Pullman, WA 99164-7411, USA; a.parrish@wsu.edu (A.N.P.); ilange@wsu.edu (I.L.); dsamec@unin.hr (D.Š.)

**Keywords:** *Equisetum*, flavonoid, galactolipid, metabolomics, phenolic acid conjugate, quantitative PCR (polymerase chain reaction), styrylpyrone

## Abstract

Three species of the genus *Equisetum* (*E. arvense, E. hyemale*, and *E. telmateia*) were selected for an analysis of chemical diversity in an ancient land plant lineage. Principal component analysis of metabolomics data obtained with above-ground shoot and below-ground rhizome extracts enabled a separation of all sample types, indicating species- and organ-specific patterns of metabolite accumulation. Follow-up efforts indicated that galactolipids, carotenoids, and flavonoid glycosides contributed positively to the separation of shoot samples, while stryrylpyrone glycosides and phenolic glycosides were the most prominent positive contributors to the separation of rhizome samples. Consistent with metabolite data, genes coding for enzymes of flavonoid and galactolipid biosynthesis were found to be expressed at elevated levels in shoot samples, whereas a putative styrylpyrone synthase gene was expressed preferentially in rhizomes. The current study builds a foundation for future endeavors to further interrogate the organ and tissue specificity of metabolism in the last living genus of a fern family that was prevalent in the forests of the late Paleozoic era.

## 1. Introduction

Members of the genus *Equisetum* are often referred to as “living fossils”, partly because they are the only extant representatives of the Equisetidae, a subclass that was once prominent—in terms of abundance, diversity (three orders comprising more than 15 genera), and size (up to 30 m tall)—in late Paleozoic forests [1]. Extant horsetails are generally divided into three lineages, the subgenera *Equisetum* (seven species), *Hippochaete* (seven species), and *Paramochaete* (one species) [2]. Horsetails are characterized by the presence of below-ground rhizomes and above-ground photosynthetic, segmented shoots. Vegetative shoots carry whorls of small leaves (microphylls) that emerge from each junction between segments (node), while unbranched fertile shoots bear a strobilus (spore-containing cone) at their tips (Figure 1A) [3]. Shoots of horsetails are coated with abrasive silicates and have, thus, been used for cleaning metal items and polishing wood crafts (hence, the common name scouring rush for *E. hyemale* L.). Records for the use of *Equisetum* in herbal remedies date back several centuries, and cell-based assays have yielded promising results; however, the evidence for clinical efficacy has remained sparse [4].

Several classes of specialized metabolites have been reported to occur in the genus *Equisetum*. The structurally related alkaloids palustrine, *N*^5^-formylpalustrine, and palustridine were identified in early phytochemical studies, during the 1930s to 1950s, as toxic constituents of *E. palustre* L. [5,6,7]. Studies performed during the early 1970s to mid-1990s focused on the identification and characterization of flavonoid glycosides and caffeic acid conjugates in shoot tissue samples from various *Equisetum* species [8,9,10,11,12,13,14,15,16,17,18,19,20,21,22]. A red pigment, the ketocarotenoid rhodoxanthin, was isolated from *E. arvense* L. shoots and characterized in the late 1970s [23]. Sterols and steroids that accumulate in *Equisetum* species were first reported in the 1980s and 1990s [24,25,26]. During the mid to late 1990s, styrylpyrone glycosides were identified for the first time as metabolites of rhizomes and gametophytes of several *Equisetum* species [27,28]. More recently, fatty acid esters of sucrose were reported to occur in *E. hyemale* [29], the coumarin herniarin was isolated from *E. debile* roxb., the unusual sesquiterpene equisetumone was found in *E. palustre*, phenolic, lignan, and sesquiterpenoid glycosides were detected in several *Equisetum* species, the occurrence of the alkaloid equisetumine was established in *E. debile*, and the major constituents of epicuticular waxes of shoots were described [29,30,31,32,33,34,35,36,37,38,39]. While steady progress is being made with identifying individual novel metabolites of *Equisetum*, few if any analyses have focused on assessing the chemical diversity across the genus [40].

The biosynthesis of specialized metabolites in *Equisetum* was first studied in the mid- 1990s. Partially purified protein fractions obtained from shoots were demonstrated to contain activities that catalyze the formation of various phenolic esters with hydroxycinnamoyl-coenzyme A as an acceptor [41]. Styrylpyrone synthase activity, catalyzing the first committed step in styrylpyrone biosynthesis, was first detected in partially purified protein fractions from cultured gametophytes [42,43]. The structure of chalcone synthase from *Equisetum*, which forms naringenin, the signature precursor of flavonoids, was reported more recently [44].

We have an ongoing interest in furthering our understanding of the metabolic diversity associated with the evolution of early land plant lineages [45,46,47,48]. In this context, we now report on a multi-omic analysis of three *Equisetum* species, studying metabolite and transcript abundance patterns in both rhizomes and shoots. Our results lay the foundation for continued research to capture the metabolic capabilities in the fern allies.

## 2. Results and Discussion

### 2.1. Experimental Design Considerations

Three species of horsetail were selected for a metabolomics experiment: *E. hyemale* subsp. *affine* (rough horsetail or scouring rush), which is native to the temperate to artic portions of North America; *E. arvense* (common horsetail), which is endemic to the arctic and temperate regions of the northern hemisphere; *Equisetum telmateia* subsp. *braunii* (Milde) Hauke (giant horsetail), which is native to western North America (Figure 1B). Both below-ground rhizomes and above-ground sterile shoots were collected from greenhouse-grown plants of each species (five biological replicates), freeze-dried, and separately ground to homogenates. Samples were extracted with 80% aqueous methanol and analyzed by nontargeted high-performance liquid chromatography–quadrupole time-of-flight mass spectrometry (HPLC–QTOF-MS). Multivariate statistical analyses were performed to assess patterns of metabolite accumulation across species and tissue types in *Equisetum*. Next-generation ribonucleic acid sequencing (RNA-Seq) data were obtained with representative samples, which enabled the identification of candidate genes involved in the biosynthesis of the major classes of specialized metabolites in *Equisetum*. The expression levels of selected genes were then determined by quantitative real-time polymerase chain reaction (qRT-PCR), once again with five biological replicates per sample type (Figure 1B).

### 2.2. Sample Differentiation Based on HPLC–QTOF-MS Data

Molecular features (detector signals representing a specific retention time and accurate mass-to-charge ratio) were extracted from HPLC–QTOF-MS raw data; the processed data sets were log-transformed and then subjected to an unsupervised principal component analysis (PCA) (Appendix A). A full separation of *Equisetum* samples (tissue types and species), with tight clustering of biological replicates for each sample type, was achieved in the first three principal components (accounting for 74.5% of data variance) (Figure 2A). The first principal component (PC1; explaining 41.2% of variance) placed samples in two well-separated groups, one comprising all rhizome samples the other encompassing all shoot samples (Figure 2B). PC2 (explaining 19.0% of variance) isolated *E. telmateia* (both rhizome and shoot) and *E. hyemale* rhizome samples from the remaining samples. *E. arvense* and *E. hyemale* shoot samples were separated from the remainder in PC3 (explaining 14.3% of variance) (Figure 2B).

The molecular features contributing the most to sample separation in PCA were then analyzed for patterns of accumulation (focusing on those with high confidence peak annotation). Galactolipids with unsaturated fatty acid side-chains, carotenoids, and flavonoid glycosides were the classes of metabolites that contributed most prominently to the PC1 placement (positive PCA scores) of *Equisetum* shoot samples (Figure 3A). Our observation that above-ground shoot samples preferentially accumulate monogalactosyldiacylglycerol (MGDG) and digalactosyldiacylglycerol (DGDG) with a high content of polyunsaturated fatty acid side-chains (compared to below-ground rhizomes) is in excellent agreement with a recent report that these bilayer-forming galactolipids are predominantly found in photosynthetic tissues of *Equisetum variegatum* Schleich. ex. Web. and *E. scirpoides* Michx. collected in the Russian permafrost zone [49]. The relatively high abundance of carotenoids in photosynthetic tissues of *Equisetum*, compared to non-photosynthetic rhizomes, is also consistent with the literature [50]. The enrichment of flavonoid glycosides in vegetative shoot samples (compared to rhizomes), as observed in the present study, is likewise in accordance with previous publications [15,27]. Styrylpyrones and styrylpyrone glycosides were the dominant contributors to the positioning of rhizome samples in PC1 (negative PCA scores) in our experiments (Figure 3B). These uncommon polyketides were previously reported to be mostly lacking in sterile shoots of *Equisetum*, which is again consistent with our findings [27,28,32]. In summary, the present work confirms the tissue specificity of galactolipid, carotenoid, flavonoid glycoside, and styrylpyrone accumulation, but this is the first demonstration of the utility of an untargeted analytical approach to profile multiple metabolite classes for sample differentiation.

The unique positioning of *E. telmateia* rhizome samples in PC2 (highly positive PCA scores) correlated with the comparatively high abundance of molecular features that could not be associated with structures of known metabolites (Figure 3C). Rhizome samples of *E. hyemale* had strong negative PCA scores in PC2 due to the contributions of molecular features with also mostly unknown identity (except for a phenolic glycoside) (Figure 3D). *E. arvense* samples were positioned uniquely in a PCA plot (highly positive PC3 scores) because of the relatively high abundance of certain flavonoid and phenolic glycosides (Figure 3E). Relatively high levels of several flavonoid glycosides and a megastigmane glucoside (derived from carotenoid breakdown) in *E. hyemale* shoot samples correlated with the strong negative PCA score in PC3 (Figure 3F). Taken together, PCA performed with our untargeted metabolomics data allowed us to differentiate both *Equisetum* species and tissue types.

### 2.3. Annotation of Metabolites That Contribute Substantially to the Separation of Samples by PCA

To ensure that the general conclusions from our PCA were based on strong analytical evidence, we employed a stepwise process to obtain high confidence peak annotations. Briefly, molecular formulas were calculated on the basis of mass-to-charge ratios and isotope patterns of the detected molecular features; these formulas were then used to search against open-source metabolite libraries (KNApSAcK, Metlin, PubChem, Spektraris, and SwissLipids). Hits returned from these searches (level 1) were further evaluated on the basis of published studies on metabolites characterized from *Equisetum* species (as available through Chemical Abstracts and Google Scholar) (level 2). The previously reported order of elution of relevant metabolites separated under comparable HPLC conditions was taken into account (level 3), and we acquired HPLC–QTOF-MS data with dozens of authentic standards (level 4 for a match of retention time (R_t_), exact mass, and relative maxima in ultraviolet/visible absorption spectra (UV/Vis)).

Phenolic acids, phenolic acid conjugates, and phenolic glycosides were among the earliest eluting analytes in our HPLC–QTOF-MS runs. Annotations with the highest confidence (level 4) were obtained for caffeoyl tartaric acid (or caftaric acid) and dicaffeoyltartaric acid (or chicoric acid), both of which were previously described as *Equisetum* metabolites [13,15,20,51]; in addition, chromatographic and mass spectral characteristics were matched with those of authentic standards (R_t_ 3.40 min, exact mass 312.0438 Da, UV/Vis 326 nm; R_t_ 12.15 min, exact mass 474.0793 Da, UV/Vis 328 nm, respectively) (Table 1). One molecular feature in our samples had characteristics consistent with an annotation as cinnamic acid (R_t_ 1.71 min, exact mass 264.1465 Da, level 2 confidence); although this metabolite is a ubiquitous constituent of plants as intermediate of the phenylpropanoid pathway, it was not previously reported in chemical analyses of *Equisetum*. To the best of our knowledge, feruloylputrescine (R_t_ 4.53 min, exact mass 148.0521 Da) and coniferylcinnamate (R_t_ 10.64 min, exact mass 330.1246 Da), two phenolic acid conjugates putatively identified in our extracts, were not previously described to occur in *Equisetum*, and the annotation in our study is, thus, tentative (level 1). Two peaks with characteristics in agreement with published phenyl glycosides were detected (level 2 evidence): equisetumoside A or B (R_t_ 7.24 min, exact mass 416.1686 Da, UV/Vis 279 nm; these metabolites are isobaric and could not be distinguished) and equisetumoside D (R_t_ 7.79 min, exact mass 414.1533 Da, UV/Vis 280 nm) (Table 1) [33,37].

Flavonoid glycosides also eluted fairly early under our HPLC–QTOF-MS conditions (6–12 min). The highest confidence annotations (level 4; match of characteristics with literature and authentic standards) were achieved for luteolin-5-*O*-glucoside (R_t_ 8.06 min, exact mass 448.1008 Da, UV/Vis 242, 342 nm), quercetin-3-*O*-glucoside (isoquercitrin; R_t_ 10.60 min, exact mass 464.0962 Da, UV/Vis 256, 354 nm), and kaempferol-3-*O*-glucoside (astragalin; R_t_ 11.78 min, exact mass 448.1015 Da, UV/Vis 264, 346 nm) (Table 1) [13,15,51,52]. Additional annotations (level 3) resulted from comparisons of characteristic HPLC parameters with those reported in the literature (kaempferol-3-*O*-sophoroside-7-*O*-glucoside (R_t_ 6.08 min, exact mass 772.2065 Da, UV/Vis 264, 344 nm), quercetin-3,7-di-*O*-glucoside (R_t_ 7.05 min, exact mass 626.1473 Da, UV/Vis 254, 348 nm), kaempferol-3,7-di-*O*-glucoside (R_t_ 8.06 min, exact mass 610.1522 Da, UV/Vis 264, 346 nm), kaempferol-3-*O*-rutinoside-7-*O*-glucoside (R_t_ 8.20 min, exact mass 756.2103 Da, UV/Vis 264, 346 nm), and kaempferol-3-*O*-sophoroside (R_t_ 9.95 min, exact mass 610.1540 Da, UV/Vis 264, 345 nm)) (Table 1) [13,15].

Styrylpyrone glycosides are signature metabolites of the *Equisetum* gametophyte and rhizomes (eluting at 6–11 min). Hispidin (6-(3,4-dihydroxystyryl)-4-hydroxy-2-pyrone) was the only representative of this class that we were able to purchase from a commercial source. Although we did not find evidence for its occurrence in *Equisetum*, the datasets acquired with the authentic standard (R_t_ 10.8 min, exact mass 246.0528 Da, UV/Vis 255, 379 nm) were important as a reference for the chromatographic and spectral properties of members of this class. Thus, on the basis of comparisons with literature reports, we tentatively identified four styrylpyrone glycosides in our samples (evidence level 3): 3-hydroxyhispidin-3,4′-di-*O*-glucoside (R_t_ 6.68 min, exact mass 586.1530 Da, UV/Vis nm), equisetumpyrone (R_t_ 8.39 min, exact mass 424.1013 Da, UV/Vis 253, 372 nm), 3′-deoxyequisetumpyrone (R_t_ 9.97 min, exact mass 408.1065 Da, UV/Vis 270, 367 nm), and 4′-*O*-methylequisetumpyrone (R_t_ 10.97 min, exact mass 438.1084 Da, UV/Vis 252, 370 nm) (Table 1) [27,28,52].

Four peaks with the typical characteristics of carotenoids were detected. Only one of these could be assigned to a structure with high confidence (level 4); an authentic standard of violaxanthin, a carotenoid with ubiquitous presence in plants [53], had the same characteristics as those of a peak in our chromatograms (R_t_ 33.83 min, exact mass 600.4177 Da, UV/Vis 415, 440, 469 nm) (Table 1). Other peaks with tentative annotation (level 1) as carotenoids (R_t_ 30.6 min, exact mass 580.3882 Da; R_t_ 31.75 min, exact mass 582.4068 Da; R_t_ 31.91 min, exact mass 616.4125 Da) did not match the exact mass values of typical plant carotenoids [53]. Debiloside B, a megastigmane glucoside (derived from carotenoid breakdown), was tentatively identified by comparison with literature data (evidence level 2) (R_t_ 8.28 min, exact mass 388.2094 Da, UV/Vis spectrum nondescript due to a lack of chromophores) (Table 1) [39].

Galactolipids were among the metabolite classes contributing to the separation of above-ground shoot samples in PCA. Two subclasses with different combinations of unsaturated fatty acids as side-chains were particularly abundant (listed in order of elution and using the shorthand nomenclature for fatty acids adopted by the International Union of Pure and Applied Chemistry): MGDG (34:6, R_t_ 33.70 min, exact mass 746.4962 Da; 36:6, R_t_ 35.02 min, exact mass 774.5274 Da; 34:4, R_t_ 35.46 min, exact mass 750.5278 Da; 36:5, R_t_ 35.86 min, exact mass 776.5443 Da; 34:3, R_t_ 36.54 min, exact mass 752.5428 Da; 36:4, R_t_ 36.72 min, exact mass 778.5575 Da; 36:2, R_t_ 37.32 min, exact mass 754.5557 Da) and DGDG (34:6, R_t_ 33.15 min, exact mass 908.5494 Da; 34:5, R_t_ 31.97 min, exact mass 910.5640 Da; 36:6, R_t_ 32.47 min, exact mass 936.5790 Da; 36:5, R_t_ 33.31 min, exact mass 938.5952 Da; 34:3, R_t_ 33.94 min, exact mass 914.60 Da; 36:4, R_t_ 34.19 min, exact mass 940.6076 Da; 34:2, R_t_ 34.77 min, exact mass 916.6117 Da). Except for MGDG-36:2 and DGDG-36:5 (level 1 evidence), authentic standards were available to ensure a level 4 annotation confidence for these lipid species (Table 1) [49].

### 2.4. Expression Patterns of Genes Putatively Involved in the Biosynthesis of Specialized Metabolites in Equisetum

Differences in the accumulation of phenolic acid conjugates, flavonoid glycosides, styrylpyrone glycosides, and galactolipids helped explain the separation of *Equisetum* samples by PCA. Interestingly, the pathways that lead to these classes of metabolites share common intermediates (Figure 4). It was, thus, of interest to evaluate if differing metabolite profiles were also reflected in gene expression patterns. As a first step to identify candidate genes for follow-up experimentation, we obtained RNA-Seq data with representative samples (*E. arvense* shoot, *E. arvense* rhizomes, *E. hyemale* shoot, *E. hyemale* rhizomes, *E. telmateia* shoot, and *E. telmateia* rhizomes) (Appendix A). We then used sequences of genes known to be involved in the biosynthesis of the above-mentioned metabolite classes in *Equisetum* and other species and searched for putative orthologs in our RNA-Seq datasets. We were particularly interested in genes that code for enzymes with functions at metabolic branchpoints, with an emphasis on those that do not occur as part of large gene families. Sequences of *Equisetum* gene candidates were previously deposited for phenylalanine ammonia lyase (PAL; general phenylpropanoid pathway; National Center for Biotechnology Information (NCBI) accession number AY803283), chalcone synthase (CHS; flavonoid pathway; AB30004), and a polyketide synthase with styrylpyrone synthase activity (annotated as *p*-coumaroyltriacetic acid synthase or CTAS; FJ443125; Colpitts, 2009) (Appendix A). Sequences with high homology to genes coding for monogalactosyldiacylglycerol synthase (MGD; galactolipid pathway) and digalactosyldiacylglycerol synthase (DGD; galactolipid pathway) in other species were also identified. Furthermore, genes encoding enzymes with housekeeping functions (β-actin and glyceraldehyde-3-phosphate dehydrogenase) were tested as references (Appendix A). The sequences of reference genes and genes of interest were then employed to design primers for qRT-PCR (Appendix A).

The transcript for CHS was dramatically more abundant in shoots when compared to rhizomes (530-, 210-, and 104-fold higher in *E. arvense*, *E. hyemale*, and *E. telmateia* samples, respectively) (Figure 5A). These results are consistent with the preferential accumulation of flavonoid glycosides in above-ground tissues. Transcript levels for MGD were also higher in shoots than in rhizomes (1.9-, 2.0-, and 2.0-fold for *E. arvense*, *E. hyemale*, and *E. telmateia*, respectively) (Figure 5B), which agreed with galactolipid accumulation patterns. The same patterns were observed for DGD (1.5-, 2.4-, and 1.8-fold shoot-to-rhizome difference in *E. arvense*, *E. hyemale*, and *E. telmateia*, respectively) (Figure 5C). The PAL transcript was expressed at lower levels in shoots than in rhizomes in *E. arvense* and *E. telmateia* (0.5- and 0.3-fold, respectively) (Figure 5D); in contrast, PAL expression was comparable in *E. hyemale* samples. PAL plays a central role in the general phenylpropanoid pathway [54], major products of which are flavonoid glycosides that accumulate primarily in shoots and styrylpyrone glycosides that are present mainly in rhizomes, but there are many other products derived from this pathway. It will remain to be investigated if the differences in PAL expression observed in above-ground and below-ground organs of *E. arvensis* and *E. telmateia* correlate with the concentrations of these metabolites. Our primers targeted the most prominent isoform of PAL expressed in shoots and rhizome samples (based on RNA-Seq data); thus, it would be of interest to investigate isoform-specific expression patterns in the future (beyond the scope of the current study). Comparable transcript abundance across samples was detected for a polyketide synthase gene (CTAS) with high homology to the bonified SPS of kava (*Piper methysticum* G. Forst) [55] in *E. arvense*; in contrast, a dramatically lower transcript abundance was observed for this transcript in shoots of *E. hyemale* and *E. telmateia* (when compared to rhizome samples) (0.2- and 0.1-fold, respectively) (Figure 5E). Interestingly, a biochemical analysis of CTAS from *E. hyemale* indicated that the enzyme could produce *p*-coumaroyltriacetic acid (a linear triketide) or triketide pyrones (cyclized) from *p*-coumaroyl-CoA, with the outcome of the reaction depending on the assay conditions [56]. It is, thus, unclear if this gene codes for an enzyme involved in styrylpyrone biosynthesis. Our RNA-Seq data revealed the presence of another polyketide synthase gene represented by contig DN52675 in *E. telmateia* rhizomes (Appendix A). qRT-PCR data provided evidence that this transcript was less abundant in shoots compared to rhizomes (0.7-, 0.01-, and 0.5-fold for *E. arvense*, *E. hyemale*, and *E. telmateia*, respectively) (Figure 5F). This pattern of expression would appear to agree with our metabolite data that showed a rhizome-specific accumulation of styrylpyrones. A full functional evaluation of this gene is beyond the scope of this study, but it would certainly be interesting to further investigate the kinetic properties of the encoded enzyme in the context of styrylpyrone biosynthesis.

## 3. Materials and Methods

### 3.1. Plant Growth

*E. arvense, E. hyemale*, and *E. telmateia* (voucher specimens deposited with the John G. Searle Herbarium of the Field Museum, Chicago, IL, USA) were maintained in a greenhouse under ambient lighting, with supplemental lighting from sodium-vapor lamps (Ethical Statement: Equisetum plants used in this study were kindly provided by the University of California Botanical Garden in Berkeley, CA, USA). The photosynthetically active radiation varied from 15 to 25 mol·m^−2^·day^−1^. Temperatures ranged between 22 and 27 °C, and the humidity was set to 70% ± 10%. Five biological replicates (separate plants) were harvested at the same time of day for below-ground rhizomes and above-ground shoots of vegetative shoots. Samples were snap-frozen in liquid nitrogen and freeze-dried (aerial parts for 5 days, rhizomes for 7 days). Lyophilized material was submerged in liquid nitrogen, homogenized using a mortar and pestle, and then stored in separate batches at −80 °C until further processing.

### 3.2. Tissue Extraction and Analysis by HPLC–QTOF-MS

Frozen tissue homogenate (30 mg per sample) was transferred to a 2 mL reaction tube and extracted with 1 mL of 80% aqueous methanol (containing 10 mg/L anthracene-9-carboxylic acid as internal standard) by vigorous shaking (VX-2500 multi-tube vortexer, VWR Scientific, South Plainfield, NY, USA) for 10 min and subsequent sonication for 20 min (FS30 ultrasonic cleaner, Fisher Scientific, Hampton, NY, USA). Following centrifugation for 10 min at 13,000× *g* (5415 microfuge, Eppendorf, Enfield, CT, USA), the supernatant was filtered through 0.22 μm polypropylene syringe filter tips, and the flow-through was collected in plastic inserts for 2 mL reaction vials. The conditions used for the separation and detection of metabolites by HPLC–QTOF-MS are the same as reported previously [48].

### 3.3. HPLC–QTOF-MS Data Processing and Statistical Analyses

Raw datasets were opened in the MassHunter Profinder version B.06.00 (build 6.0.0625.0) software package (Agilent Technologies, Santa Clara, CA, USA), and molecular features were obtained using the batch recursive feature extraction algorithm. Binning and alignment tolerances were set to 10% + 20 s for the retention time, 10 ppm + 2 mDa for the mass accuracy, and 0.0025 *m*/*z* + 5.0 ppm for the isotope grouping space tolerance. Additional parameters that were considered for feature extraction were quasi-molecular ions and adducts ([M + H]^+^, [M + Na]^+^, [M + K]^+^, [M + NH_4_]^+^), dimers, neutral losses (H_2_O, H_3_PO_4_, C_6_H_10_O_5_ (glucose), C_12_H_20_O_9_ (rutinose), C_12_H_20_O_10_ (sophorose), C_6_H_10_O_4_ (rhamnose), and C_5_H_8_O_4_ (xylose)), absolute peak height ≥2000 counts, and occurrence required in a minimum of four of the five replicates of each sample type. These preprocessing steps generated 848 molecular features, for which data were exported to an Excel spreadsheet. Additional exclusion criteria for molecular features were as follow: relative standard deviation of mass accuracy ≥5.0 ppm; percentage relative standard deviation returned as “NaN” (not a number) or an empty cell; an unacceptably close accurate mass and retention time (±0.010 *m*/*z* and ±0.02 min; screened as duplicates); if it was a fragment. This additional filtering returned 544 remaining molecular features. Peak areas of molecular features for each sample were normalized on the basis of the sample weight and the peak area of the internal standard (molecular features without a peak area were filled in with a nominal value of two). Preprocessed datasets were imported into RStudio version 1.4.1717 [57] running *R* version 4.1.1 [58] and subjected to log_10_ transformation, autoscaling, and centering. Data dimensionality reduction was performed using unsupervised PCA with the prcomp function in *R*. The *R* packages factoextra, ggplot2, and pca3d were employed for visualization and figure generation.

The MassHunter Qualitative Analysis software version B.07.00 Service Pack 1 (build 7.0.7024.29) (Agilent Technologies, Santa Clara, CA, USA) was employed to generate molecular formulas with the “common organic molecule isotope model” setting. Molecular formulas were then used in searches against public databases, including PubChem [59], KNApSAcK [60], and Spektraris [46,61]. Annotations were also evaluated against the available literature on metabolites previously described to occur in horsetail samples. For high confidence annotations, retention time and accurate mass data for molecular features were compared with those of authentic standards. HPLC–QTOF-MS data and meta data for the current study were submitted to the National Metabolomics Data Repository (Project ID PR001223) [62].

### 3.4. RNA Extraction, RNA-Seq, and Data Processing

Total RNA was extracted using the RNeasy Mini kit (Qiagen, Hilden, Germany) according to the manufacturer’s instructions. To eliminate genomic DNA, RNA (1–2 μg) was treated with DNase I (1 unit of enzyme per μg of RNA) and then processed with the RNeasy MinElute kit (Qiagen, Valencia, CA, USA). The quality of mRNA samples was tested by taking RNA Integrity Number measurements with a 2100 Bioanalyzer (Agilent Technologies, Santa Clara, CA, USA). The TruSeq RNA Sample Preparation kit (Illumina, San Diego, CA, USA) was used to generate mRNA-focused libraries, which were then subjected to transcriptome sequencing with the HiSeq 3000 platform (Illumina, San Diego, CA, USA) to produce 101 bp paired-end reads. After assessing the read quality with FastQ, erroneous kmers, adapter sequences, and low-quality reads were removed, and sequence reads were assembled with Trinity (v 2.6.5) [63,64]. Short reads were uploaded to NCBI’s Short Read Archive (BioProject ID PRJNA340020). Expression levels (expressed as transcripts per kilobase million) were calculated by RSEM (v 1.2.22) and bowtie (v 2.0.0). Annotations were generated using Trinotate (v 3.0.2).

### 3.5. RNA Extraction, First-Strand cDNA Synthesis, qPCR, and Data Processing

Frozen plant material was homogenized under liquid nitrogen using a mortar and pestle. RNA was extracted with the Nucleo Spin RNA Plant and Fungi kit (Macherey-Nagel, Allentown, PA, USA) according to the manufacturer’s instructions. The quality of the isolated DNase-free RNA was evaluated by gel electrophoresis and UV spectroscopy. First-strand cDNA from each replicate was synthesized using the Maxima H minus reverse transcriptase (Fisher Scientific, Waltham, MA, USA) with equal amounts (1 µg) of RNA for each sample. In a 10 μL quantitative PCR reaction, concentrations were adjusted to 300 nM (primers), 1× iTaq Power SYBR Green PCR Master Mix (Bio-Rad, Hercules, CA, USA), and 10× diluted first-strand cDNA as template (primer sequences provided in Appendix A). Reactions were performed in a 96-well optical plate at 95 °C for 10 min, followed by 40 cycles of 95 °C for 15 s and 60 °C for 10 min in a CFX Connect RT System (Bio-Rad, Hercules, CA, USA). Fluorescence intensities of three independent measurements (technical replicates) were normalized against the carboxyrhodamine reference dye (ThermoFisher, Waltham, MA, USA). Genes coding for β-actin and glyceraldehyde 3-phosphate dehydrogenase were tested as constitutively expressed endogenous controls, with the former giving the most consistent expression levels across samples (used as reference gene). The RDML-LinRegPCR tool was employed to calculate Cq and qPCR efficiency values from raw data according to the reporting guidelines of the RDML consortium [65,66]. Relative fold-difference levels between shoot and rhizome samples were calculated using a mathematical model that adjusts for qPCR efficiency and crossing point deviation [67]. The *p*-values were obtained with a two-tiered Student’s *t*-test (t.test function) in Microsoft Excel (pairwise comparisons of above-ground and below-ground samples).

## 4. Conclusions

Principal component analysis of metabolomics data obtained with above-ground shoot and below-ground rhizome extracts of three different greenhouse-grown *Equisetum* species enabled a separation of all sample types. Shoot samples were separated from rhizome samples due to the higher accumulation of galactolipids, carotenoids, and flavonoid glycosides, while rhizome samples were enriched in stryrylpyrone glycosides and phenolic glycosides. Consistent with metabolite profiles, shoot samples had elevated levels of genes coding for enzymes of flavonoid and galactolipid biosynthesis, while a putative styrylpyrone synthase gene was expressed preferentially in rhizomes. We recognize that our data only provide a snapshot of metabolite and gene expression patterns under a specific set of controlled greenhouse conditions; thus, it would be of great interest for future efforts to assess chemical diversity patterns in different *Equisetum* species across biomes.

## Figures and Tables

**Figure 1 metabolites-12-00403-f001:**
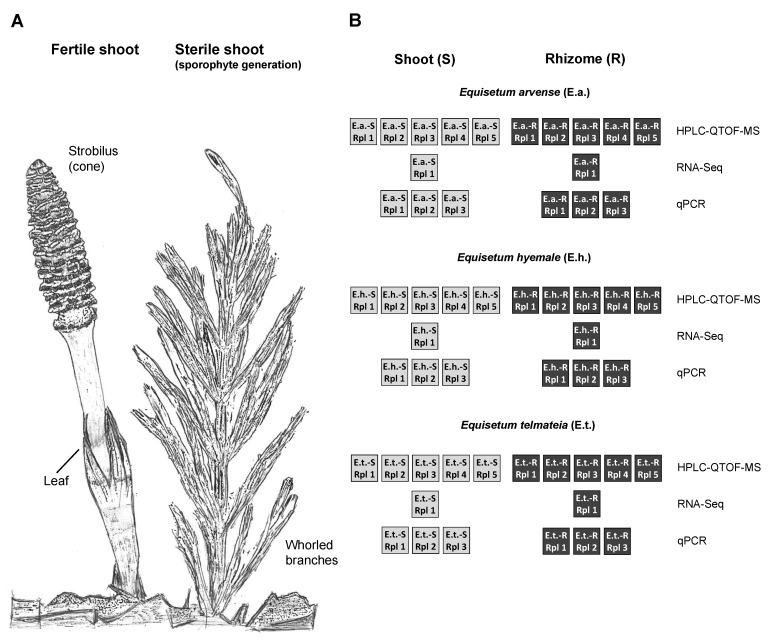
Multi-omics analysis of *Equisetum* samples. (**A**) Sketch of a fertile and sterile shoot. (**B**) Experimental design. Abbreviations: E.a., *Equisetum arvense*; E.h., *Equisetum hyemale*; E.t., *Equisetum telmateia*; HPLC–QTOF-MS, high-performance liquid–quadrupole time-of-flight mass spectrometry; RT-qPCR, quantitative real-time polymerase chain reaction; R, rhizome; RNA-Seq, next-generation ribonucleic acid sequencing; Rpl, replicate; S, shoot.

**Figure 2 metabolites-12-00403-f002:**
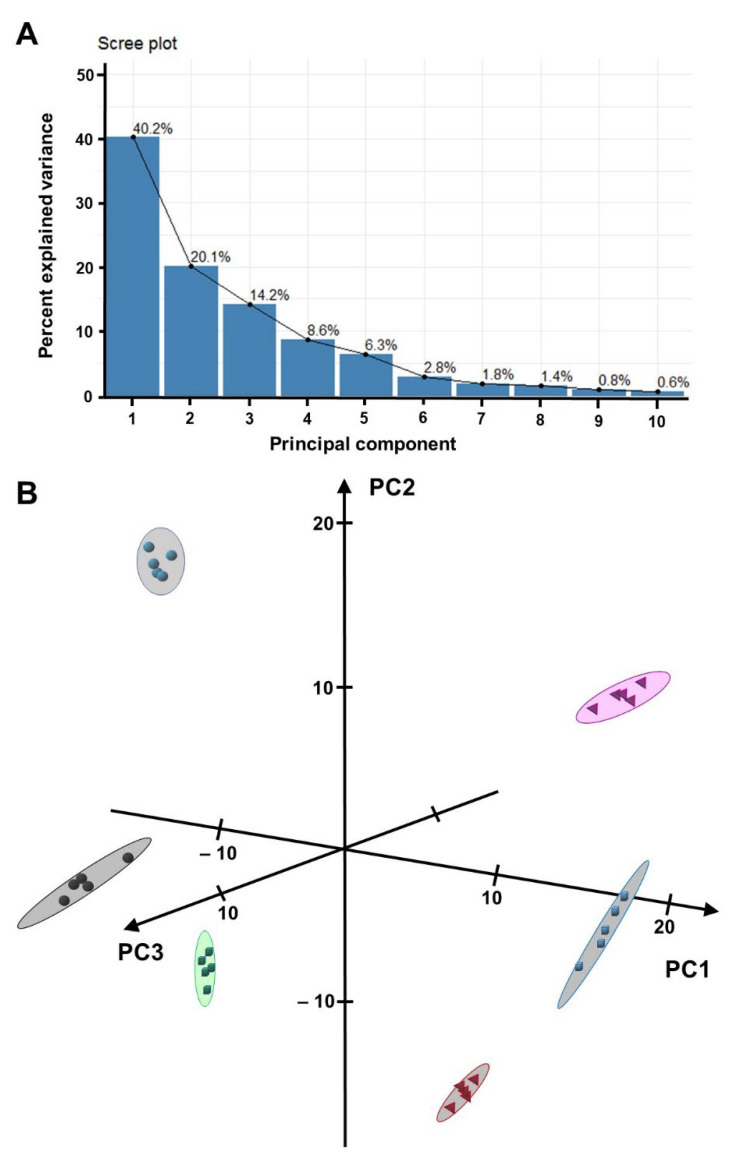
Separation of *Equisetum* sample groups based on a principal component analysis (PCA) of metabolomics data. (**A**) Scree plot indicating that the first three principal components (PCs) explain 74% of variance across datasets. (**B**) Three-dimensional PCA plot visualizing the separation of samples from *Equisetum arvense* shoots (red pyramids), *Equisetum arvense* rhizomes (black spheres), *Equisetum hyemale* shoots blue cubes), *Equisetum hyemale* rhizomes (green cubes), *Equisetum telmateia* shoots (magenta pyramids), and *Equisetum telmateia* rhizomes (teal spheres).

**Figure 3 metabolites-12-00403-f003:**
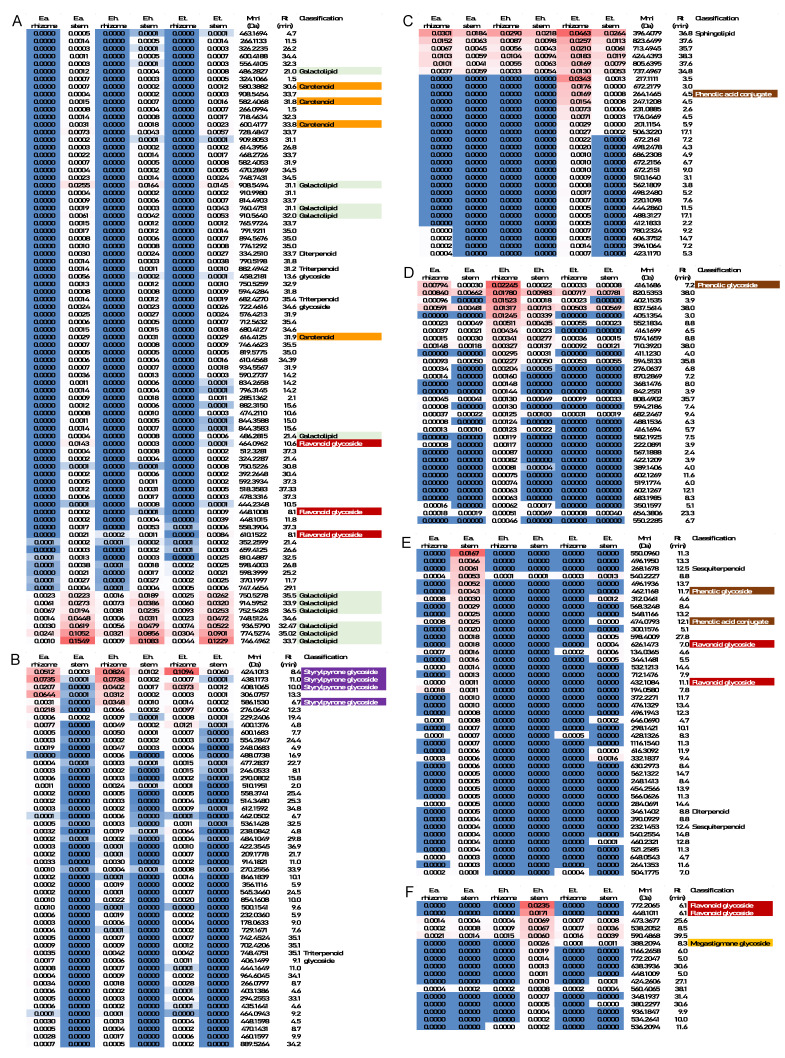
Classification of molecular features contributing most strongly to *Equisetum* sample group separation in PCA, organized by patterns of accumulation: (A) highly positive PC1 scores; (**B**) strongly negative PC1 scores; (**C**) highly positive PC2 scores; (**D**) strongly negative PC2 scores; (**E**) highly positive PC3 scores; (**F**) strongly negative PC3 scores. Abbreviations: E.a., *Equisetum arvense*; E.h., *Equisetum hyemale*; E.t., *Equisetum telmateia*; *m*/*z*, mass-to-charge ratio; Rt, retention time.

**Figure 4 metabolites-12-00403-f004:**
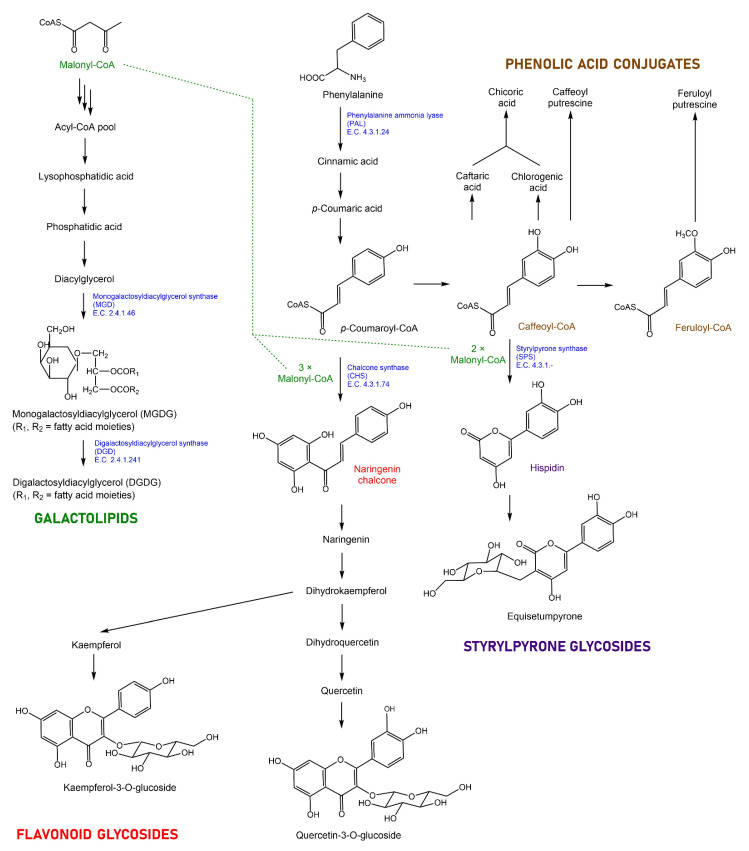
Representative structures of the main classes of specialized metabolites detected in *Equisetum* samples and outline of the relevant biochemical pathways. Enzymes selected for follow-up research (by assessing the expression levels of the corresponding genes) are shown in blue font.

**Figure 5 metabolites-12-00403-f005:**
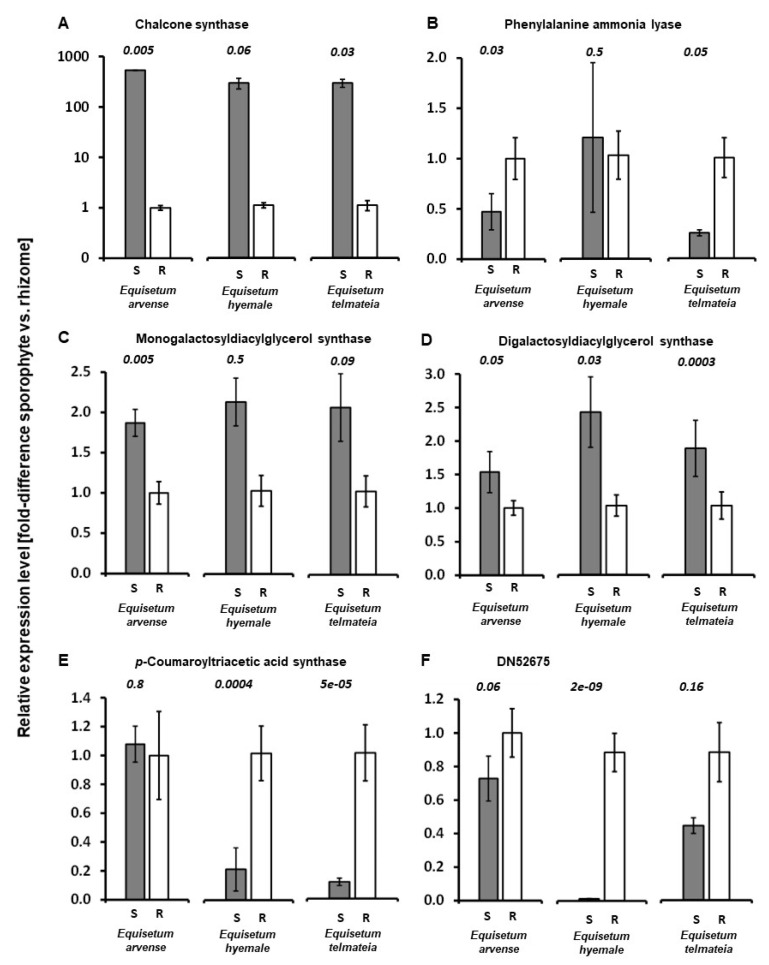
RT-qPCR analysis of expression patterns of selected genes (expressed as fold-change of shoot (gray column) versus rhizomes (white column); standard errors shown as bars): (**A**) chalcone synthase; (**B**) phenylalanine ammonia lyase; (**C**) monogalactosyldiacylglycerol synthase; (**D**) digalactosyldiacylglycerol synthase; (**E**) *p*-coumaroyltriacetic acid synthase; (**F**) transcript corresponding to the sequence of contig DN52675 (obtained during assembly of RNA-Seq data for *E. telmateia* rhizomes). Abbreviations: S, shoot; R, rhizome. The *p*-values obtained with a two-tiered Student’s *t*-test are shown in italics (comparison shoot versus rhizome for each species).

**Table 1 metabolites-12-00403-t001:** Annotation of HPLC–QTOF-MS peaks.

Accurate Mass—Time Tag	Monoisotopic Mass (Measured/Calculated)	Δppm	Molecular Formula	MS (ESI-Positive) (* Most Abundant)	Tentative Annotation	References; Evidence Level, Standard Source
** *Phenolic acids, phenolic acid conjugates, and phenolic glycosides* **						
BML-LCMS-19- 1.71–148.0521	148.0521/148.0524	2.29	C9H8O2	[M + H]^+^ 149.0594 [M + NH_4_]^+^ 166.0867 *	Cinnamic acid	[13,15]; 2
BML-LCMS-19- 3.40–312.0438	312.0438/312.0481	2.98	C13H12O9	[M + Na]^+^ 335.0371 *	Caffeoyl tartaric acid (caftaric acid)	[13,15]; 4, Sigma Aldrich 88,656
BML-LCMS-19- 4.53–264.1465	264.1465/264.1474	1.34	C14H20N2O3	[M + H]^+^ 265.1552 * [M + Na]^+^ 287.1366	Feruloylputrescine	No reference; 1
BML-LCMS-19- 7.24–416.1686	416.1686/416.1682	0.62	C19H28O10	[M + Na]^+^ 439.1579 *	Equisetumoside A or B	[33]; (2)
BML-LCMS-19- 7.79–414.1533	414.1533/414.1526	1.38	C19H26O10	[M + Na]^+^ 437.1426 *	Equisetumoside D	[37]; (2)
BML-LCMS-19- 10.64–310.1246	310.1246/310.1205	0.55	C19H18O4	[M + H]^+^ 311.1281 *	Coniferylcinnamate	No reference; 1
BML-LCMS-19- 12.15–474.0793	474.0793/474.0798	0.60	C22H18O12	[M + Na]^+^ 497.0690 *	Dicaffeoyltartaric acid (chicoric acid)	[13,15,20,51]; (4) Cayman 24,960
** *Flavonoid glycosides* **						
BML-LCMS-19- 6.08–772.2065	772.2065/772.2062	0.93	C33H40O21	[M + H]^+^ 773.2147 *	Kaempferol 3-*O*-sophoroside-7-*O*-glucoside	[13,15]; 3
BML-LCMS-19- 7.05–626.1473	626.1473/626.1483	1.93	C27H30O17	[M + H]^+^ 627.1545 *	Quercetin-3,7-di-*O*-glucoside	[13,15]; 3
BML-LCMS-19- 8.06–448.1008	448.1008/448.1006	0.37	C21H20O11	[M + H]^+^ 449.1079 *	Luteolin-5-*O*-glucoside	[13,15,51,52]; 4, Sigma Aldrich 1,370,837
BML-LCMS-19- 8.06–610.1522	610.1522/610.1534	0.37	C27H30O16	[M + H]^+^ 611.1614 * [M + Na]^+^ 633.1432	Kaempferol-3,7-*O*-di-glucoside	[13,15]; 3
BML-LCMS-18- 8.20–756.2103	756.2103/756.2113	0.75	C33H40O20	[M + H]^+^ 757.2181 *	Kaempferol-3-*O*-rutinoside-7-*O*-glucoside	[13,15]; 3
BML-LCMS-19- 9.95–610.1540	610.1540/610.1534	1.23	C27H30O16	[M + H]^+^ 611.1622 [M + Na]^+^ 633.1438 *	Kaempferol-3-*O*-sophoroside	[13,15]; 3
BML-LCMS-19- 10.60–464.0962	464.0962/464.0955	1.02	C21H20O12	[M + H]^+^ 465.1036 * [M + Na]^+^ 487.0849	Quercetin 3-glucoside (isoquercitrin)	[13,15,51,52]; 4, Sigma Aldrich 00140585
BML-LCMS-19- 11.78–448.1015	448.1015/448.1006	0.01	C21H20O11	[M + Na]^+^ 471.0900 *	Kaempferol-3-*O*-glucoside (astragalin)	[13,15,51,52]; 4, Cayman 25,060
** *Styrylpyrone glycosides* **						
BML-LCMS-19- 6.68–586.1530	586.1530/586.1534	0.51	C25H30O16	[M + H]^+^ 587.2013 *	3-Hydroxyhispidin-3,4’-di-*O*-glucoside	[52]; 3
BML-LCMS-19- 8.39–424.1013	424.1013/424.1006	1.37	C19H20O11	[M + H]^+^ 425.1085 * [M + Na]^+^ 447.0903	Equisetumpyrone	[28,52]; 3
BML-LCMS-19- 9.97–408.1065	408.1065/408.1056	1.31	C19H20O10	[M + H]^+^ 409.1136 * [M + Na]^+^ 431.0957	3’-Deoxyequisetumpyrone	[27]; 3
BML-LCMS-19- 10.97–438.1173	438.1173/438.1162	1.63	C20H22O11	[M + H]^+^ 439.1245 * [M + Na]^+^ 461.1064	4’-*O*-Methylequisetumpyrone	[27]; 3
** *Carotenoids and apocarotenoids* **						
BML-LC-MS-18- 8.28–388.2094	388.2094/388.2097	0.72	C19H32O8	[M + Na]^+^ 411.1991 *	Debiloside B	[39]; 2
BML-LCMS-19- 30.60–580.3882	580.3882/580.3916	1.88	C40H52O3	[M + H]^+^ 581.3978 *	Carotenoid	No reference; 1
BML-LMS-19- 31.75–582.4068	582.4068/582.4073	0.03	C40H54O3	[M + H]^+^ 583.4142 *	Carotenoid	No reference; 1
BML-LCMS-19- 31.91–616.4125	616.4125/616.4128	0.05	C40H56O5	[M + Na]^+^ 639.4013 *	Carotenoid	No reference; 1
BML-LCMS-19- 33.83–600.4177	600.4177/600.4179	0.63	C40H56O4	[M + H]^+^ 601.4246 *	Violaxanthin	[53]; 4, Sigma Aldrich 52,444
** *Lipids* **						
BML-LCMS-19- 21.01–486.2827	486.2827/486.2829	2.55	C25H42O9	[M + NH_4_]^+^ 504.3157 [M + Na]^+^ 509.2718 *	16:3-Glycosylmonoacylglycerol	No reference; 1
BML-LCMS-19- 21.36–486.2815	486.2815/486.2829	2.83	C25H42O9	[M + NH_4_]^+^ 504.3157 [M + Na]^+^ 509.2714 * [M + K]^+^ 525.2461	16:3-Glycosylmonoacylglycerol	No reference; 1
BML-LCMS-19- 31.15–908.5494	908.5494/908.5497	0.79	C49H80O15	[M + NH_4_]^+^ 926.6327 [M + Na]^+^ 931.5382 *	Digalactosyldiacylglycerol (34:6)	[49]; 4, Avanti 840,524 (mix)
BML-LCMS-19- 31.97–910.5640	910.5640/910.5654	1.55	C49H82O15	[M + Na]^+^ 933.5536 * [M + K]^+^ 949.5267	Digalactosyldiacylglycerol (34:5)	[49]; 4, Avanti 840,524 (mix)
BML-LCMS-19- 32.47–936.5790	936.5790/936.5810	1.36	C51H84O15	[M + NH_4_]^+^ 954.6140 [M + Na]^+^ 959.5708 *	Digalactosyldiacylglycerol (36:6)	[49]; 4, Avanti 840,524 (mix)
BML-LCMS-19- 33.31–938.5952	938.5952/938.5967	0.15	C51H86O15	[M + Na]^+^ 961.5872 *	Digalactosyldiacylglycerol (36:5)	No reference; 1
BML-LCMS-19- 33.70–746.4962	746.4962/746.4969	0.03	C43H70O10	[M + NH_4_]^+^ 764.5311 * [M + Na]^+^ 769.4859	Monogalactosyldiacylglycerol (34:6)	[49]; 4, Avanti 840,523 (mix)
BML-LCMS-19- 33.94–914.5952	914.5952/914.5967	0.71	C49H86O15	[M + NH_4_]^+^ 932.6303 [M + Na]^+^ 937.5848 *	Digalactosyldiacylglycerol (34:3)	[49]; 4, Avanti 840,524 (mix)
BML-LCMS-19- 34.19–940.6076	940.6076/940.6123	0.19	C51H88O15	[M + Na]^+^ 963.6026 *	Digalactosyldiacylglycerol (36:4)	[49]; 4, Avanti 840,524 (mix)
BML-LCMS-19- 34.77–916.6117	916.6117/916.6123	0.94	C49H88O15	[M + NH_4_]^+^ 934.6454 [M + Na]^+^ 939.6016 *	Digalactosyldiacylglycerol (34:2)	[49]; 4, Avanti 840,524 (mix)
BML-LC-MS-18- 35.02–774.5274	774.5274/774.5282	0.34	C45H74O10	[M + NH_4_]^+^ 792.5619 * [M + Na]^+^ 797.5171	Monogalactosyldiacylglycerol (36:6)	[49]; 4, Avanti 840,523 (mix)
BML-LC-MS-18- 35.46–750.5278	750.5278/750.5282	0.66	C43H74O10	[M + NH_4_]^+^ 768.5588 [M + Na]^+^ 773.5169 *	Monogalactosyldiacylglycerol (34:4)	[49]; 4, Avanti 840,523 (mix)
BML-LC-MS-18- 35.86–776.5443	776.5443/776.5438	0.61	C45H76O10	[M + Na]^+^ 799.5325 *	Monogalactosyldiacylglycerol (36:5)	[49]; 4, Avanti 840,523 (mix)
BML-LC-MS-18- 36.54–752.5428	752.5428/752.5438	1.04	C43H76O10	[M + NH4]^+^ 770.5775 [M + Na]^+^ 775.5325 *	Monogalactosyldiacylglycerol (34:3)	[49]; 4, Avanti 840,523 (mix)
BML-LC-MS-18- 36.72–778.5575	778.5575/778.5595	1.13	C45H78O10	[M + H]^+^ 779.5910 [M + NH_4_]^+^ 796.5930 [M + Na]^+^ 801.5480 *	Monogalactosyldiacylglycerol (36:4)	[49]; 4, Avanti 840,523 (mix)
BML-LC-MS-18- 37.32–754.5557	754.5557/754.5595	1.70	C43H78O10	[M + NH_4_]^+^ 772.5921 [M + Na]^+^ 777.5483 *	Monogalactosyldiacylglycerol (36:2)	No reference; 1

## Data Availability

HPLC–QTOF-MS datasets was uploaded to the NIH Common Fund’s National Metabolomics Data Repository (NMDR) website (Project ID PR001223) [62]. RNA-Seq data are available at NCBI’s Short Read Archive (BioProject ID PRJNA340020). Processed data are included in the figures, tables, and supplementary materials of this manuscript.

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
