# Peer review of "Differential Accumulation of Metabolites and Transcripts Related to Flavonoid, Styrylpyrone, and Galactolipid Biosynthesis in Equisetum Species and Tissue Types"

_metabolites, 2022, doi:10.3390/metabo12050403_

Round 1

Reviewer 1 Report

I have carefully read MS which was submitted for consideration in the Metabolites (MDPI). The topic of the work is undoubtedly very important. In this study, the authors report on a multi-omic analysis of three species of horsetail, i.e. E. arvense, E. hyemale, and E. telmateia by examining metabolites and transcripts abundance patterns in rhizomes and shoots. Their results lay the foundations for further research into capturing the metabolic capabilities of Polypodiopsida. This article is very relevant and interesting, and also is very important to understand the evolutionary diversity in an ancient land plant lineage. It should be noted that the genus Equisetum is a "living fossil", the only living taxon of the entire Equisetidae subclass that has survived over 100 million, since the late Paleozoic.

This manuscript is in general well written, logically structured, well-illustrated and easy to understand. It also addresses a subject that is of great interest in the scientific community. The title clearly describes the contents of the paper. The abstract is well written. It encapsulates the entire study (a bit of introduction, aim, result and outcome).  The introduction is well written as it gives a good background of the research in question. Also, the aim of the study is evident in the beginning and concluding parts. I believe that the Materials and Methods section is well structured and scientifically sound. The results are well presented, figures and tables are correct. Literature reviews in the discussion section of the manuscript are very professional.

Minor:

Please consider correcting Figure A (Figure 1), the drawing of the horsetail is poorly visible.

Author Response

We thank the reviewer for the positive evaluation.  A new image for Figure 1A is now included in the revised version of the manuscript.

Reviewer 2 Report

The manuscript entitled “Differential accumulation of Metabolites and Transcripts related to Flavonoid, Styrylpyrone, and Galactolipid Biosynthesis in Equisetum Species and Tissue types” by Parrish et al analyzed three species of the genus Equisetum (E. arvense, E.hyemale, and E. telmateia) about the chemical diversity. 

PCA analysis showed that differences in the accumulation of flavonoid glycosides, styrylpyrone glycosides, and galactolipids explained the separation of Equisetum. Consistent with metabolite data, genes coding for enzymes of flavonoid and galactolipid biosynthesis were found to be expressed at elevated levels in shoot samples, whereas a putative styrylpyrone synthase gene was expressed preferentially in rhizomes. The authors further conducted qRT-PCR to investigate the expression pattern of selected genes coding enzymes (chalcone synthase, phenylalanine ammonia lyase, monogalactosyldiacylglycerol synthase, digalactosyldiacylglycerol synthase, p-coumaroyltriacetic acid synthase, DN52675).

Overall, the method used in the study is thorough. Conclusions are appropriate, and supported by the data. Statistical analysis is provided within the manuscript. The whole study is sound, and I recommend accepting it. 

Author Response

We thank the reviewer for the positive evaluation.

Reviewer 3 Report

The manuscript “Differential Accumulation of Metabolites and Transcripts Related to Flavonoid, Styrylpyrone, and Galactolipid Biosynthesis in Equisetum Species and Tissue Types”, intends to report on a multi-omic analysis of three Equisetum species, studying metabolite and transcript abundance patterns in both rhizomes and shoots. Moreover, this study builds a foundation for future endeavours to further interrogate the organ- and tissue-specificity of metabolism in the last living genus of a fern family that was prevalent in the forests of the late Paleozoic era.

The manuscript addresses an interesting topic, it is well organized and clearly describes the methods and the results. Therefore, in my opinion, the manuscript may be recommended for publication after the authors added a study conclusion

Author Response

We thank the reviewer for the positive evaluation.  Based on their comments, a conclusion paragraph has been added to the revised version of the manuscript.

Reviewer 4 Report

The manuscript deals with accumulation of different metabolites differentially according to the organs of three Equisetum species (E. arvense, E. hyemale, and E. telmateia) by using a multi-omics approch.

The manuscript seems interesting and meets fully the expectations of Metabolites.

The used design is sound. The sampling was done adequatly and the methods of analyses are currents.

The manuscript is well written and the methodology clearly described.

Some parts of the manuscript are boring to read. Nevertheless, this is due to omics results.

few points should be addressed.

1- statistical analyses are not described.

2- the manuscript is based on one experiment which is very limited. Therefore authors should "mitigate" their conclusion. Indeed the used experiment was done under controlled conditions which not reproductible.

3- the most important concern is the lack of conclusion.

Author Response

We thank the reviewer for the suggestions to improve our manuscript. 

Comment:

1- statistical analyses are not described.

Response:

In the revised version of the manuscript, we are providing the following information about statistical analyses:

Metabolomics data

Peak areas of molecular features for each sample were normalized based on sample weight and the peak area of the internal standard (molecular features without a peak area were filled in with a nominal value of two).  Pre-processed data sets were imported into RStudio version 1.4.1717 (https://www.rstudio.com/) running R version 4.1.1 (https://cran.r-project.org/) and subjected to log10 transformation, autoscaling, and centering.  Data dimensionality reduction was performed using unsupervised PCA with the prcomp function in R.  The R packages factoextra, ggplot2, and pca3d were employed for visualization and figure generation.  

qPCR data

P-values were obtained with a two-tiered student t-test (t.test function) in Microsoft Excel (pairwise comparisons of above-ground and below-ground samples). 

Comment:

2- the manuscript is based on one experiment which is very limited. Therefore authors should "mitigate" their conclusion. Indeed the used experiment was done under controlled conditions which not reproductible.

Response:

We are not quite sure which conclusions should be "mitigated".  Our experimental design includes multiple experimental replicates, which are based on independent tissue harvests.  This means that each extraction and analysis is a separate experiment.

We are also not entirely sure why data obtained with plants growth under highly controlled, well-described conditions would not be reproducible.  As multiple independent biological replicates indicate, the data is very reproducible.  Maybe the reviewer is eluding to the fact that plants grown under greenhouse conditions may not allow us to capture the diversity of metabolites occurring in plants collected outdoors?  This would certainly be correct.  We thus added a sentence to acknowledge this fact to the new Conclusions paragraph:

"Our data provide insights into the occurrence of organ- specific metabolic pathways, and it would now be of great interest to assess chemical diversity patterns in different Equisetum species across biomes. "

Comment:

3- the most important concern is the lack of conclusion.

Response:

In the revised version of the manuscript, we have added the following Conclusions paragraph:

"Principal Component Analysis of metabolomics data obtained with above-ground shoot and below-ground rhizome extracts of three different greenhouse-grown Equisetum species enabled a separation of all sample types.  Shoot samples were separated from rhizome samples due to the higher accumulation of galactolipids, carotenoids and flavonoid glycosides, while rhizome samples were enriched in stryrylpyrone glycosides and phenolic glycosides.  Consistent with metabolite profiles, shoot samples had elevated levels of genes coding for enzymes of flavonoid and galactolipid biosynthesis, while a putative styrylpyrone synthase gene was expressed preferentially in rhizomes.  Our data provide insights into the occurrence of organ- specific metabolic pathways, and it would now be of great interest to assess chemical diversity patterns in different Equisetum species across biomes."

Round 2

Reviewer 4 Report

First I would like to thank authors for the improvements done on the manuscript.

Nevertheless, I still maintain that the manuscript is based on one experiment which is very limited. Therefore authors should "mitigate" their conclusion. Indeed the used experiment was done under controlled conditions which can not reproductible.

Indeed, even in greenhouse, the external climatic conditions impact the internal controlled conditions. Moreover, these interne conditions are also dependent on the material provider. I cannot accept the rebuttal answer of authors.

I think, It would therefore be more appropriate to say that these results were obtained under their conditions.

Author Response

The reviewer suggests that we should emphasize the fast that the data reflect results obtained under specific growing conditions.  To address this comment, we changed the last sentence of the Conclusions as follows:

"We recognize that our data only provides a snapshot of metabolite and gene expression patterns under a specific set of controlled greenhouse conditions; it would thus be of great interest for future efforts to assess chemical diversity patterns in different Equisetum species across biomes."